# Motion Analysis of the Extensor Carpi Ulnaris in Triangular Fibrocartilage Complex Injury Using Ultrasonography Images

**DOI:** 10.3390/s22218216

**Published:** 2022-10-27

**Authors:** Shuya Tanaka, Atsuyuki Inui, Yutaka Mifune, Hanako Nishimoto, Tomoya Yoshikawa, Issei Shinohara, Takahiro Furukawa, Tatsuo Kato, Masaya Kusunose, Ryosuke Kuroda

**Affiliations:** Department of Orthopaedic Surgery, Graduate School of Medicine, Kobe University, 5-2, Kusunoki-cho 7, Chuo-ku, Kobe-shi 650-0017, Japan

**Keywords:** extensor carpi ulnaris, triangular fibrocartilage complex injury, ultrasonography, particle image velocimetry, ulnar deviation, radial deviation, long-axis view

## Abstract

The subsheath of the extensor carpi ulnaris (ECU) tendon, a component of the triangular fibrocartilage complex (TFCC), is particularly important as it dynamically stabilizes the distal radioulnar joint. However, the relationship between TFCC injury and ECU dynamics remains unclear. This study aimed to analyze ECU movement and morphology using ultrasonography (US) images. Twenty wrists of patients with TFCC injury, who underwent TFCC repair, were included in the injury group, and 20 wrists of healthy volunteers were in the control group. For static image analysis, curvature and linearity ratios of the ECU in US long-axis images captured during radioulnar deviation were analyzed. For dynamic analysis of the ECU, the wrist was moved from radial deviation to ulnar deviation at a constant speed, and the velocity of the tendon was analyzed using particle image velocimetry. The static analysis showed that the ECU tendon was more curved in ulnar deviation in the injury group than in the control group, and the dynamic analysis showed that only vertical velocity toward the deep side during ulnar deviation was higher in the injury group. These results suggest that TFCC injury caused ECU curvature during ulnar deviation and increased the vertical velocity of the ECU during wrist deviation.

## 1. Introduction

The triangular fibrocartilage complex (TFCC) is an important component involved in distal radioulnar joint (DRUJ) stability [1,2,3]. TFCC injury causes ulnar-sided wrist pain, clicks, and a decrease in a power grip, and it can result in remarkable wrist disability and impaired hand function [3]. The TFCC consists of the articular disc, dorsal and volar radioulnar ligaments, meniscus homologue, ulnar collateral ligament, and sheath of the extensor carpi ulnaris (ECU) [4]. The sheath of the ECU tendon is particularly important, as it functions as a dynamic stabilizer of the DRUJ [5]. Tang et al. created cadaveric models of TFCC avulsion by releasing the TFCC, which replicated a Palmer-type IB injury of the TFCC, and they reported that changes in ECU excursion and bowstringing of the ECU tendon could be significantly affected by the avulsion of the TFCC [6].

The ulnocarpal stress test is used to evaluate TFCC injury during ulnar deviation (UD) with pronation or supination [7]. A previous magnetic resonance imaging (MRI) analysis showed wavy deformities on the articular disc during UD in patients with horizontal tears inside the TFCC [8]. A previous study reported that TFCC injury potentially contributes to abnormal loading and force on the ECU during wrist deviation and that it affects the ECU tendon morphological features, such as curvature or tortuosity. However, the relationship between TFCC injury and ECU morphology in vivo has not been clarified in previous studies.

In recent years, ultrasonography (US) has been increasingly used for diagnosis and treatment because it is a low-cost and minimally invasive modality that enables dynamic evaluation in vivo. To date, there are a few US evaluation reports of ECU tendon dislocation analyzed in the short-axis view [9,10,11] and one report that analyzed ECU movement in the long-axis view, which compared a TFCC injury group with a control group [12]. The study showed that the magnitude velocity of the ECU in TFCC injury patients was significantly higher than that in healthy volunteers.

Particle image velocimetry (PIV) is a quantitative flow visualization tool developed to measure fluid velocity in the field of fluid mechanics. Several studies have used the PIV method for the US evaluation of musculoskeletal diseases [13,14]. PIV has been used to detect a decrease in the gliding properties of the vastus lateralis and subcutaneous tissue in patients with residual pain after trochanteric fracture surgery. Previous studies have reported on the high reliability and reproducibility of the PIV method [12,13].

We hypothesized that the ECU tendon morphological features (such as curvature or tortuosity) and movement features (such as bowstringing or excursion) in TFCC injury wrists have changed compared to normal wrists. Thus, the present study aimed to clarify how TFCC injury affects the morphology and movement of the ECU tendon using US images in a long-axis view. To evaluate ECU morphology, we aimed to analyze the curvature and linearity of the ECU tendon using static US images captured during UD and radial deviation (RD). To evaluate the ECU movement, we aimed to analyze the velocity of the tendon movement for each vector using PIV as a dynamic analysis tool. This was the first study to analyze the morphology and movement of the ECU in TFCC injury wrists in vivo.

## 2. Materials and Methods

Twenty healthy volunteers (12 men and eight women; mean age, 31.2 years) (control group) and 20 patients with Palmer type 1B TFCC injuries (14 men and six women; mean age, 27.1 years) (injury group), who had undergone surgical treatment between April 2020 and March 2022, were included. TFCC injury was diagnosed using MRI and intraoperative arthroscopic findings. Patients with TFCC injuries other than Palmer type 1B injuries were excluded from the study. The sample size was determined by power analysis based on data from the pilot study [12], using G*Power 3.1. Prior sample size calculations showed that a difference in velocimetry of ECU was detectable in the two groups with a sample size of 36 participants (18 participants in each group), using the t-test (effect size = 0.99, α = 0.05, power = 0.8).

We identified the ECU in US long-axis images of the TFCC according to Wu et al.’s protocol [15] (Figure 1). Participants actively moved their wrists from 15° of RD (Max RD) to 25° of UD (Max UD) at a 60-rounds-per-minute rhythm paced by a metronome in a pronation and dorsiflexion neutral position, and we recorded videos at 30 frames per second, as reported previously [12].

For a static evaluation, we calculated the curvature and linearity ratios of the ECU with the help of static images captured in the Max RD and Max UD positions, respectively, using Image J software (USA National Institutes of Health (NIH)). The curvature and linearity ratios were defined according to a previous study on river meandering [16]. Since we could not find a method to evaluate the degree of meandering of single-curved structures in medical literature, we decided to use the evaluation method for river meandering. The curvature ratio was defined as the amplitude divided by the curvature length, and the linearity ratio was defined as the tendon width divided by the curvature width (Figure 2). Amplitude was defined as the vertical distance between the dorsal border of the ECU at the level of the ulnar styloid process and at the level of the articular disc where the ECU tendon was located on the deepest side. Curvature length was defined as the distance of the straight line between the ulnar styloid process and triquetrum. Tendon width was defined as the mean of the tendon width measured at the following points: at the level of (1) the ulnar styloid process, (2) articular disc, and (3) triquetrum. Curvature width was defined as the sum of the amplitude and tendon width.

For a dynamic evaluation, the U component, which comprises the longitudinal velocity of the ECU, with a positive value indicating a movement toward the proximal direction, and the V component, which comprises the vertical velocity, with a positive value indicating a movement toward the deep direction, during each phase of UD and RD, were calculated through the PIV method using PIV Lab, Version 2.36, with MATLAB (Mathworks, Natick, MA, USA) (Figure 3). The ECU tendon between the ulnar styloid process and triquetrum was set as the region of interest in PIV.

All values are expressed as mean ± standard deviation (SD). Comparisons of all the parameters between each group were analyzed using the *t*-test, and *p* < 0.05 was considered indicative of a statistically significant difference. The Excel add-in statistical software package (Ekuseru-Toukei 2015; Social Survey Research Information Co., Ltd., Tokyo, Japan) was used to perform the statistical analysis.

## 3. Results

The curvature ratios were 0.075 ± 0.036 in Max UD and 0.079 ± 0.042 in Max RD in the control group and 0.108 ± 0.057 in Max UD and 0.088 ± 0.046 in Max RD in the injury group. The linearity ratios were 0.65 ± 0.12 in Max UD and 0.62 ± 0.15 in Max RD in the control group and 0.58 ± 0.12 in Max UD and 0.62 ± 0.11 in Max RD in the injury group (Table 1, Figure 4). The curvature ratio in Max UD was significantly higher and the linearity ratio in Max UD was significantly lower in the injury group than those in the control group (*p* = 0.021 and *p* = 0.048, respectively). The ratios in Max RD were not significantly different between the two groups (*p* = 0.27 and *p* = 0.50, respectively). In the injury group, the curvature ratio in Max UD was significantly higher than that in Max RD, and the linearity ratio in Max UD was significantly lower than that in Max RD (*p* = 0.013 and *p* = 0.034, respectively).

The PIV analysis showed that the U component was 0.223 ± 0.501 mm/s during UD and −0.330 ± 0.478 mm/s during RD in the control group and 0.218 ± 0.719 mm/s during UD and −0.530 ± 1.09 mm/s during RD in the injury group, while the V component was −0.116 ± 0.44 mm/s during UD and 0.075 ± 0.35 mm/s during RD in the control group and 0.283 ± 0.22 mm/s during UD and −0.235 ± 0.25 mm/s during RD in the injury group. While there was no significant difference in the U component between the two groups, the V component during both UD (*p* = 0.005) and RD (*p* = 0.007) (Table 2, Figure 5) exhibited significant differences. The positive and negative values of the V component were reversed between the groups during RD and UD (Figure 6).

## 4. Discussion

In this study, we analyzed how the ECU behaved in TFCC injury, both statically and dynamically. In the injury group, the ECU was more curved in Max UD than in Max RD, while in the control group, there were no significant differences between Max RD and Max UD. In the dynamic evaluation using PIV, a significant difference was observed in the vertical movement of the ECU. On the other hand, there was no difference in the longitudinal movement of the ECU between the two groups. In the injury group, the ECU moved toward the deep side during UD and toward the superficial side during RD; these directions were opposite to those in the control group, and the velocity was higher than that in the control group. These results indicated that TFCC injury caused ECU curvature during UD and increased the vertical velocity of the ECU during wrist deviation.

A previous US study revealed that DRUJ instability, expressed as the dorsal translation ratio of the ulnar head in patients with complete rupture of the TFCC, was significantly higher in injured wrists than in uninjured wrists [17]. To date, several reports on ECU dynamics in cadavers and in US short-axis view images have been published. The subsheath of the ECU tendon is a part of the TFCC, and attaches from the dorsal surface of base of the ulnar styloid to the dorsal side of the triquetrum. The ECU floor includes fibers of the ulnar collateral ligament, which contributes to DRUJ stability [5], and the DRUJ is stabilized during RD through tension in the ECU subsheath [18,19]. Campbell et al. reported that in the forearm supination position, the ECU tendon exits the sixth compartment at an angle of approximately 30° in the coronal plane, and tension on the ECU subsheath and retinaculum was higher during forearm supination combined with wrist flexion and UD [20]. A cadaveric biomechanical study reported that the ECU tendon exhibited a 30% increase in excursion during wrist extension after releasing the TFCC from its attachment to the distal ulna [6]. Only cadaveric studies have analyzed the relationship between TFCC injury and ECU dynamics, and a US analysis for evaluating ECU and TFCC injury in vivo has not yet been conducted.

The present study showed that the ECU tendon was more curved and less straight in the Max UD position in the injury group than in the Max RD position in the injury group and in both positions in the control group. In all cases, the tendon had a single curve with convexity on the deep side at the level of the articular disk; thus, the results indicated that the apex of the curve of the tendon in the injury group was located on the deeper side during Max UD. This was similar to the movement of the articular disk, which also migrated to the deeper side in the UD position in the TFCC injury group [12].

We used the PIV method for a dynamic evaluation of the ECU; this method can track soft tissues in US images by mimicking the small hyperechoic dots inside the tendon as tracers. Several previous studies have shown that PIV has high reliability and reproducibility [12,13]. On using PIV, the horizontal and vertical gliding movements of the ECU can be evaluated independently. In the present study, while there was no difference in longitudinal movement between the two groups, the vertical movement velocity was significantly higher during UD and significantly lower during RD in the injury group. In this measurement system, a positive value of the V component indicated faster movement toward the deep direction, while a negative value of the V component indicated a movement toward the superficial direction. Interestingly, the positive and negative values were reversed between the groups during wrist deviation. Shinohara et al. [12] also reported that the V component of the articular disc moved in different vectors between the control and injury groups. This similarity in findings supports the idea that the ECU works together with the articular disc as a part of the DRUJ stabilizer, and TFCC injury may contribute to the abnormal mobility of the ECU as well as of the articular disk.

This study has several limitations. First, only Palmer type 1B injuries were included in this study. TFCC injuries other than type 1B injuries were not included; therefore, the results of this study cannot be generalized to all TFCC injuries. Second, we evaluated tendon dynamics during only radial-ulnar deviation with forearm pronation. Images in the US long-axis view could not be obtained precisely with wrist flexion–extension and forearm supination because the limb position changed during the US examination, and it was difficult to fix the US probe to the wrist joint in the same manner. Although the results showed a significant difference even during wrist deviation, further studies are needed to evaluate other limb positions, especially forearm supination. Third, the sample size of this study was not large because only patients with Palmer type 1B injuries, who underwent surgical treatment, were included. It is still difficult to precisely detect the type of TFCC injury, even with MRI or US images. We recommend that further studies be conducted in the future, as this would help in diagnosing and treating patients appropriately.

## 5. Conclusions

In summary, we performed static and dynamic analyses of the ECU tendon in order to compare patients with TFCC injury and healthy controls. In the long-axis view of the US images, tendon curvature was enlarged in the injury group. On analyzing the dynamics of the ECU tendon, the velocity of the vertical movement during UD was higher toward the deep side in the injury group. These changes may be caused by TFCC injuries and could be one of the reasons for wrist pain. The results also suggest that an evaluation of ECU morphology and dynamics through US may be a useful method for diagnosing TFCC injury.

## Figures and Tables

**Figure 1 sensors-22-08216-f001:**
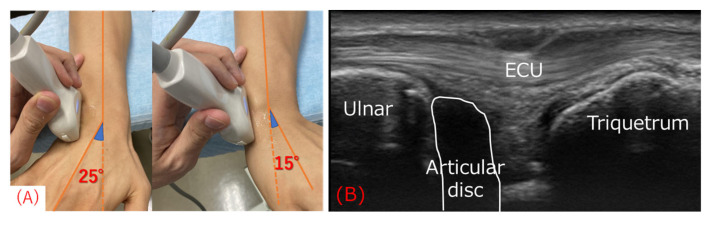
Ultrasonography (US) procedure and image in the long-axis view. (**A**) A US transducer was placed on the dorsal ulnar side of the wrist. Participants actively moved their wrists from 15º of radial deviation to 25º of ulnar deviation. (**B**) US image of the extensor carpi ulnaris (ECU) tendon at the triangular fibrocartilage complex level acquired by the procedure shown in (**A**). The ulnar styloid process was below the ECU tendon on the left side and the triquetrum on the right side.

**Figure 2 sensors-22-08216-f002:**
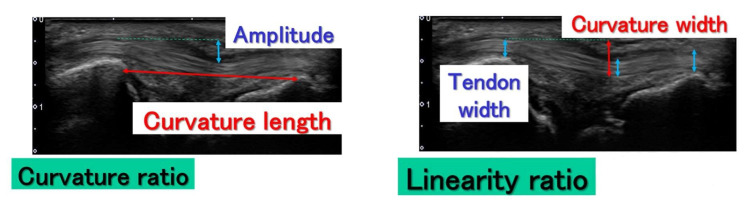
Schemas of how the curvature and linearity ratios were calculated. The curvature ratio was defined as the amplitude divided by the curvature length. The linearity ratio was defined as the curvature width divided by tendon width. The amplitude was defined as the vertical distance of the dorsal border of the extensor carpi ulnaris tendon between the level of the ulnar styloid process and location of the tendon on the deepest side. Curvature length was the distance between the ulnar styloid process and triquetrum.

**Figure 3 sensors-22-08216-f003:**
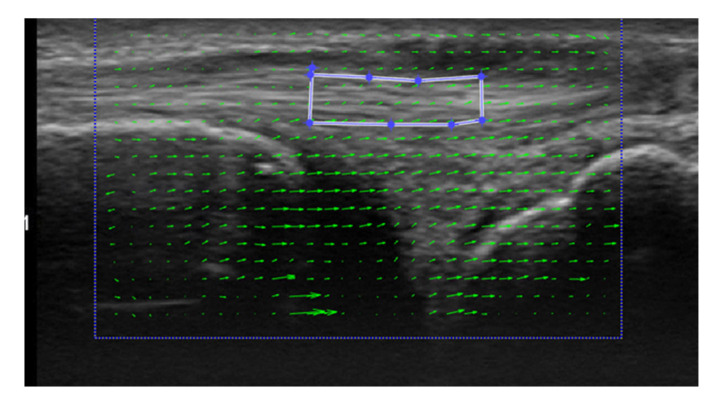
Analysis of extensor carpi ulnaris (ECU) velocity using particle image velocimetry. The green arrows indicate the directions of fluid movement. The smaller blue box indicates the regions of interest where we analyzed the ECU tendon at the level of the articular disc.

**Figure 4 sensors-22-08216-f004:**
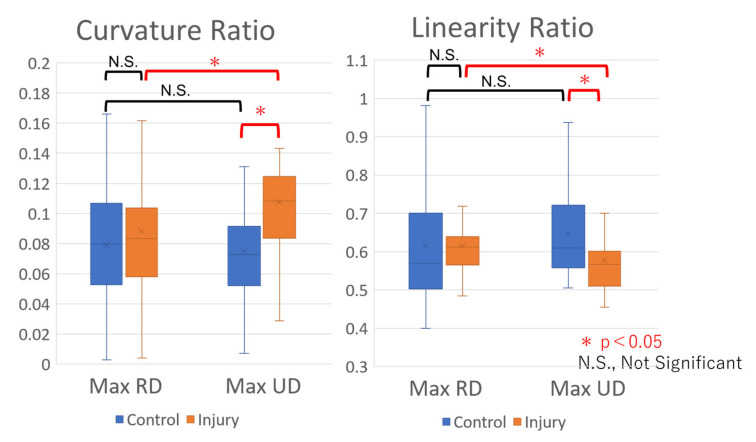
Curvature and linearity ratios in the study groups. The curvature ratio in maximum ulnar deviation (Max UD) was significantly higher and linearity ratio in Max UD was significantly lower in the injury group than those in Max UD in the control group and those in maximum radial deviation (Max RD) in the injury group.

**Figure 5 sensors-22-08216-f005:**
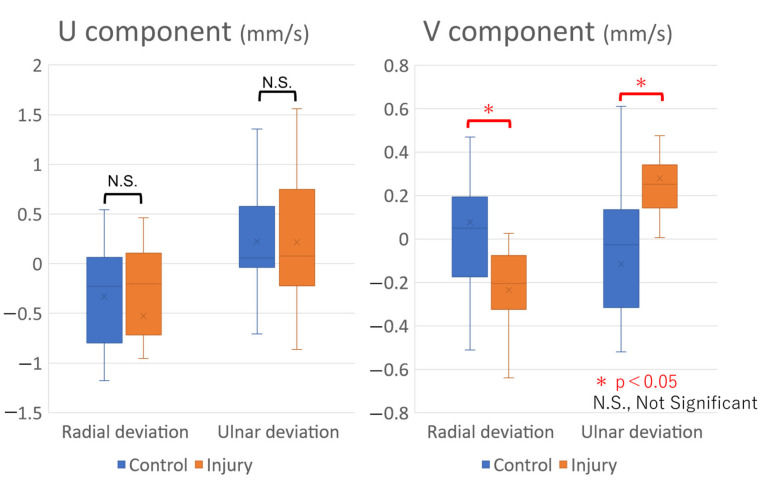
U and V components in the study groups. The U component was not significantly different between the two groups. The V component in the injury group was significantly higher than that in the control group during ulnar deviation.

**Figure 6 sensors-22-08216-f006:**
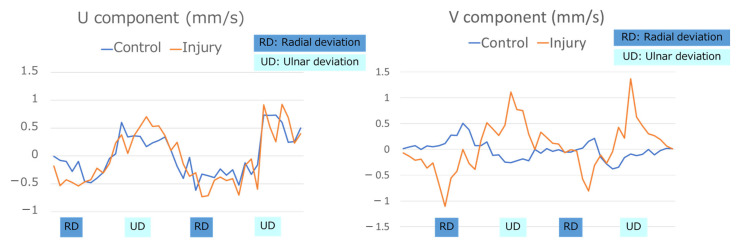
Analysis of the V and U components in the groups. The V component values revealed that the tendon moved toward the deep side during ulnar deviation (UD) and toward the superficial side during radial deviation (RD) in the injury group, while opposite findings and smaller values were seen in the control group. The U component did not reveal any differences between the two groups.

**Table 1 sensors-22-08216-t001:** The results of the curvature ratio and the linearity ratio.

	Control Group	Injury Group	*p* Value
Curvature Ratio			
Max UD	0.075 (0.036)	0.108 (0.057)	0.021
Max RD	0.079 (0.042)	0.088 (0.046)	0.27
*p* value	0.27	0.013	
Linearity Ratio			
Max UD	0.65 (0.12)	0.58 (0.12)	0.048
Max RD	0.62 (0.15)	0.62 (0.11)	0.50
*p* value	0.071	0.034	

Data were presented as mean value (standard deviation). Abbreviations: Max UD, maximum ulnar deviation; Max RD, maximum radial deviation.

**Table 2 sensors-22-08216-t002:** The results of the U component and the V component.

	Control Group	Injury Group	*p* Value
U component			
Radial deviation	−0.330 (0.478)	−0.530 (1.09)	0.30
Ulnar deviation	0.223 (0.501)	0.218 (0.719)	0.50
V component			
Radial deviation	0.075 (0.35)	−0.235 (0.25)	0.005
Ulnar deviation	−0.116 (0.44)	0.283 (0.22)	0.007

Data were presented as mean value (standard deviation).

## Data Availability

The data presented in this study are available upon request from the corresponding author. The data are not publicly available because of confidentiality concerns.

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
