# Peer review of "Motion Analysis of the Extensor Carpi Ulnaris in Triangular Fibrocartilage Complex Injury Using Ultrasonography Images"

_sensors, 2022, doi:10.3390/s22218216_

Round 1
Reviewer 1 Report
The last paragraph of introduction refers to conclusions. It should be removed, because conclusions must be after the presentation of results.
Figure 2 is cumplsy it includes graphical description of some of the dimensions used in the text as well as formulas and definitions. I suggest remove the formulas from the figure and put it as real formulas and use a formal definition of the dimensions that can be both in the caption of the figure and the main text.
The use of curvature and linearity rations from river meandering it is surprising. It would be useful for the readers an explanation on the motives the authors took these definitions as relevant.
The main results from the measurements would be easier to review (and follow in the lecture) if included in a table instead of just as part of a paragraph (lines 129-131)
Line 211. The correct way for citing would be Shinohara et al. [12] also reported that the V component of the articular disc moved in 211 different vectors between the control and injury groups
Reviewer 2 Report
1. The introduction part needs to be better grounded theoretically in terms of the specialized literature.
2. Reading the purpose of the study carefully: Thus, the present study aimed to examine how TFCC injury affects the morphology and movement of the ECU tendon using US images in the long-axis view. To evaluate ECU morphology, we aimed to analyze the curvature and linearity of the ECU tendon using static US images captured during UD and radial deviation (RD). To evaluate the ECU movement, we aimed to analyze the velocity of the tendon movement for each vector using PIV as a dynamic analysis tool, I kindly ask you to rephrase and highlight the purpose of this paper much better.
3. The following sentence: Our main conclusions were that in TFCC injury, tendon curvature was enlarged in long-axis US images, and the velocity of the vertical movement during UD was higher toward the deep side. An evaluation of ECU morphology and dynamics through US may be a useful method for diagnosing TFCC injury, is not relevant to the introduction because it is a conclusion, so kindly remove it.
4. Please specify precisely the novelties of this study and inset them at the end of the introduction so we can see what is new in this research.
5. If possible, please enter the sample size calculation.
6. I want to mention that I have included the article to analyze the similarity coefficient in the Plagiarism CheckerX software, version 6.0.11, and please perform the following word reformulations that are in bold and italics:
· The floor of the ECU tendon sheath is a part of the TFCC, and it connects the dorsal aspect of the ulnar styloid base to the dorsal surface of the triquetrum.
· Campbell et al. reported that the ECU tendon exits the sixth compartment at an angle of approximately 30° in the coronal plane in the forearm supination position, and tension on the ECU retinaculum and subsheath was greater during activities involving forearm supination combined with wrist flexion and UD [20].
Round 2
Reviewer 2 Report
The authors have adequately made all the requested changes. Congratulations on your work!